# Damage Accumulation Phenomena in Multilayer (TiAlCrSiY)N/(TiAlCr)N, Monolayer (TiAlCrSiY)N Coatings and Silicon upon Deformation by Cyclic Nanoindentation

**DOI:** 10.3390/nano12081312

**Published:** 2022-04-11

**Authors:** Anatoly I. Kovalev, Vladimir O. Vakhrushev, Ben D. Beake, Egor P. Konovalov, Dmitry L. Wainstein, Stanislav A. Dmitrievskii, German S. Fox-Rabinovich, Stephen Veldhuis

**Affiliations:** 1State Scientific Centre, I.P. Bardin Central Research Institute for Ferrous Metallurgy, 23/9 bdg, 2, Radio Str., 105005 Moscow, Russia; gareq1211@gmail.com (V.O.V.); filin.capmer@gmail.com (E.P.K.); d_wainstein@sprg.ru (D.L.W.); info@sprg.ru (S.A.D.); 2Surface Phenomena Researches Group, LLC, Staropimenovsky Lane, 6, bdg. 1, off. 4, 127006 Moscow, Russia; 3Micro Materials Ltd., Willow House, Yale Business Village, Ellice Way, Wrexham LL13 7YL, UK; ben@micromaterials.co.uk; 4Department of Mechanical Engineering (JHE-316), McMaster University, 1280 Main Street West, Hamilton, Ontario, ON L8S 4L7, Canada; gfox@mcmaster.ca (G.S.F.-R.); veldhu@mcmaster.ca (S.V.)

**Keywords:** micro-impact fatigue, wear hard coatings, study state cutting, fractal analysis

## Abstract

The micromechanism of the low-cycle fatigue of mono- and multilayer PVD coatings on cutting tools was investigated. Multilayer nanolaminate (TiAlCrSiY)N/(TiAlCr)N and monolayer (TiAlCrSiY)N PVD coatings were deposited on the cemented carbide ball nose end mills. Low-cycle fatigue resistance was studied using the cyclic nanoindentation technique. The obtained results were compared with the behaviour of the polycrystalline silicon reference sample. The fractal analysis of time-resolved curves for indenter penetration depth demonstrated regularities of damage accumulation in the coatings at the early stage of wear. The difference in low-cycle fatigue of the brittle silicon and nitride wear-resistant coatings is shown. It is demonstrated that when distinguished from the single layer (TiAlCrSiY)N coating, the nucleation and growth of microcracks in the multilayer (TiAlCrSiY)N/(TiAlCr)N coating is accompanied by acts of microplastic deformation providing a higher fracture toughness of the multilayer nanolaminate (TiAlCrSiY)N/(TiAlCr)N.

## 1. Introduction

Simple and complex ion-plasma PVD coatings based on titanium and aluminium nitrides are widely used in the metalworking industry to improve the wear resistance of cutting tools. There are several concepts for designing such coatings. The first one is based on the permanent increasing of the hardness of the nitrides. Such coatings are aimed on the passive wear resistance during cutting and they are destroyed due to damage accumulation [1]. Chemical and phase composition of the second group of hard coatings is selected in a way that provides favourable conditions for their tribo-oxidation when protective oxide films are dynamically formed on the surface of the cutting tool [2]. These amorphous oxide films have a thickness of several nanometres. They radically reduce the friction and thermal conductivity coefficients in the wear region [3]. In this way, such coatings are adapting themselves to severe tribological conditions manifesting to the greatest extent in conditions of high-speed dry cutting.

Nanolaminate multilayer coatings for cutting tools have been proposed in the last decade as a more promising alternative to traditional monolayer coatings. Multilayer coatings show high hardness, crack resistance, and toughness when compared with monolayer coatings. The general idea of the surface engineered, multilayer PVD coatings is to design their composition and architecture that provides adaptive spatio-temporal behaviour of the coating [3,4]. This generation of coatings provides the following: the highest possible adhesion to the substrate; the ability of the coating layer to accumulate and dissipate energy simultaneously that is typical for nanostructures; and high surface protection/lubrication with an additional objective of better wear behaviour in a highly loaded contact with a workpiece and a chip.

The main advantage of multilayer coatings is associated with obstruction of the transboundary propagation of microcracks. Ultimately, the cutting tool coatings break down, but we can extend their operational resource by controlling their architecture and physico-chemical transformations in the cutting zone. In recent years, the mechanisms of degradation of cutting tool coatings have been studied by microstructural methods. However, these very detailed studies have discussed superhard coatings on late wear stages. The coatings had a weak interlayer adhesive strength [5] or were multilayer ceramic–metallic or nitride–metallic coatings TiN/Ti characterized by a sharp difference in the elastic-plastic properties of individual layers [6]. The wear of hard coatings has a multi-stage mechanism. Obviously, the phase composition of the wear products of PVD coatings is also changed during various stages of cutting.

The running-in stage is the most aggressive in terms of its external impact on the cutting tool. It largely determines the further behaviour of the tool during cutting. Protection against damage accumulation is the first and the most significant function of the coating at this stage. Indeed, features of a defective structure formation in wear-resistant coatings at the initial stage of wear are not studied sufficiently regardless of the fact that this stage determines the further service of the cutting tool. Nanolaminate multilayer composites deformed in the wear zone are complicated objects for structural studies because the accumulation of crystal lattice defects including their influence on tool wear during operation is an unusual task. The aim of this work was the investigation of damage accumulation in multilayer adaptive coatings at the running-in stage and their influence on subsequent stages of wear.

Nanoindentation impact testing is one of the representative methods of mechanical testing providing rich information about the elastic–plastic properties of materials. This method allows studying physical and mechanical properties at the nanoscale down to interatomic distances inaccessible to common mechanical testing techniques such as standard Vickers, Berkovich, Knoop, and Rockwell hardness tests [7,8,9]. Until now, theoretical and experimental studies in this area have looked for ways to improve the accuracy in determining the strength, toughness, and ductility values for various materials basing on indentation data [10,11,12]. Nanoindentation is one of the most significant methods to measure mechanical properties of surface layers and thin films in submicron regions [13].

This method allows measuring the alteration in micromechanical properties of the nanolaminate coating in course of cyclic deformation. Statistical processing of time series allows us to analyse the influence of previous events on subsequent ones. The successful experience of such application of cyclic nanoindentation was demonstrated in a study on the wear of PVD multilayer coatings [14]. In this work, damage accumulation in solid, single layer (TiAlCrSiY)N and multilayer (TiAlCrSiY)N/(TiAlCr)N PVD coatings at the initial wear stage was investigated. These results were compared with the destruction of hard and brittle silicon.

## 2. Materials and Methods

The multilayer Ti_0.2_Al_0.55_Cr_0.2_Si_0.03_Y_0.02_N/Ti_0.25_Al_0.65_Cr_0.1_N and monolayer Ti_0.2_Al_0.55_Cr_0.2_Si_0.03_Y_0.02_N coatings were deposited using Ti_0.2_Al_0.55_Cr_0.2_Si_0.03_Y_0.02_ and Ti_0.25_Al_0.65_Cr_0.1_ targets that were correspondingly fabricated by a powder metallurgical process on cemented carbide ball nose end mills WC-Co substrate in an R&D-type hybrid PVD coater (Kobe Steel Ltd., Koube, Japan) using a plasma-enhanced arc source. The multilayer coating had 30 bilayers. WC-Co samples were heated up to about 500 °C and cleaned by Ar ion etching. During the PVD process, an Ar-N_2_ gas mixture was fed to the chamber at a pressure of 2.7 Pa with nitrogen partial pressure of 1.3 Pa. The arc source was operated at 100 A for a 100 mm diameter × 16 mm thick target. The other deposition parameters were bias voltage 100 V and substrate rotation 5 rpm. The thickness of the coating studied was around 3 μm for the film characterization and micro-scale impact testing. The coating has a nanocrystalline multi-layered microstructure with alternating nanolayers periods of 20–30 nm [15,16] and a hardness of 30 GPa measured using the nanoindentation method at room temperature and 28 GPa at 500 °C [17]. Under micromechanical testing these coatings were compared with polycrystalline Si. Micromechanical characteristics were determined using a Micro Materials NanoTest System setup. A trigonal diamond pyramid with the angle of 65.3° penetrated the sample with coating at the loads of 25 mN, 30 mN, 40 mN, and 50 mN. The cyclic load and unloading with the period of 4 s were repeated many times for 5 min. The penetration depth vs. the cyclic loading time was registered automatically. Prior to indentation tests, the calibration of the indenter tip was carried out employing a fused silica sample. Cyclic nanoindentation loading were carried out on three different samples of Si, monolayer, and multilayer coated systems and the penetration depth versus time was recorded continuously at a constant indentation rate of 0.05 s^−1^.

These studies allowed for modelling the behaviour of the material at low-cycle fatigue. In the micro-impact test there is a quasi-static indentation before the first actual impact. The on-load indentation depth (h_0_) associated with this is recorded and used to confirm that the depth zero is measured correctly and that the test did not impact an anomalous region of the surface. Subsequently, the probe depth is recorded “on-load” for every impact. A detailed description of cyclic nanoindentation technique is given in [2]. In our case, all micro-impact tests were carried out by one diamond indenter to exclude the influence of probe geometry [18].

## 3. Results and Discussion

Cutting tools are operating under low-cycle fatigue conditions. At the initial stage of cutting tool wear, elastic–plastic deformation is observed, accompanied by strain hardening and active nucleation, and accumulation of crystal lattice imperfections. The micro-cracking completes these processes of plastic deformation. In this regard, the resistance of the coating to low-cycle fatigue can affect the operational resource of the cutting tool.

Indicative displacement/penetration vs. time curves for four different indentation maximum loads ranging from 25 mN to 50 mN are shown in Figure 1. During the first few impacts, the depth increases rapidly, gradually slowing to approach a plateau where the depth is almost unchanged with each successive impact. The monolayer coating compared to the multilayer one is characterised by lower resistance to cyclic loading already at the earliest stage of the testing. Moreover, already in the first 30 s, the indenter penetration in the single layer coating was about 1000 nm, and penetration depth increases rapidly. A sharp jump on the curve means the formation of forked microcracks propagating to a big depth. In the multilayer coating, plastic deformation was developed gradually with some incubation time, and after 50 s the maximal penetration depth did not exceed 280 nm at the bug jump. This is comparable to the propagation of microdefects through six bilayers. During the incubation time, at almost 20 s from the start of the experiment, up to 50 s, the swings were about 30–50 nm of the penetration depth corresponding to the nucleation and propagation of cracks in one layer. After 50 s, the depth of penetration of the indenter into the monolayer coating was bigger than for the multilayer one. This means a higher level of damage accumulation at this stage of the test in the monolayer coating.

Comparable results of Si nanoimpact with spherical indenter were presented in [19].

For silicon reference samples, the results of cyclic nanoindentation differ significantly from the nitride coatings behaviour. Silicon is an extremely brittle material. It is known that depending on the load during nanoindentation, a series of phase transitions occurs in silicon. In the literature, such effects are associated with the formation of new phases, in particular, Si–II phase with an increase in the load and Si–III/Si–XII phases at the reset [19].

The increase of maximal load to 40 and 50 mN radically changes nanoindentation curves probably associated with phase transitions already initiated at the initial stage of cyclic testing because it correlates with dependences of such phase transformations on the load and loading rate described in [20]. After a short incubation period, the elastoplastic deformation in this brittle material develops, accompanied by the formation of a sufficient amount of dislocation sources forming initial microcracks, which are then merged into main ones. Nanoindentation curves demonstrate discontinuities of a spontaneous rise in the indenter penetration depth. In this case, the characteristic time of such a jump is much shorter than the experiment sampling time equal to 0.05 s. This means that Si–II → Si–III/Si–XII phase transition is accompanied by a rapid expansion of the crystalline lattice volume in a new phase of structural transformations, which leads to peak stress growth exceeding the critical ones for the formation of brittle cracks.

These discontinuities known in the literature as pop-ins and pop-outs [21] manifest themselves as sudden steps crack propagation in both loading and unloading stages. It is comprehensible that the occurrence of a pop-in is strongly connected with free length of crack propagation. The length of free propagation for such microfractures depends on the crystalline structure of different silicon polymorphic phases. A significant differentiation in jumps of the indenter penetration on these curves occurs due to variations in free propagation length of brittle microcracks. As we see in silicon, microcracks can unite and propagate over considerable distances comparable with pop-in jumps.

Conversion of test results to the dependence of contact stress from test time allows for a better understanding of the development of microstructural defects. These contact stresses could be calculated using well-known equations for hardness but one should take into account the presence of both elastic and plastic components of deformation while the microhardness values refer only to the plastic deformation of the sample. When assessing the microhardness of a heterophase layered nanocomposite, the microhardness of a bilayer is calculated taking into account the volume fraction of phases [12]. Assuming that elastoplastic properties of layers in a TiAlCrSiYN/TiAlCrN multilayer coating do not differ significantly, the contact stress σ can be determined from the generally accepted expression:σ = P_max_/A,(1)
where A is the actual projected contact area indenter with material at maximal load P_max_.

In modern nanoindentation systems, the hardness is defined without the optical control of final residual imprint. A as defined by area function is directly determined from the “load-displacement curve” [22]. A dependence of hardness and contact stress in our case on the indenter penetration depth h can be represented as:σ = 0.00387 P_max_/h^2^,(2)
where h is the penetration depth. It is clear that σ includes components of elastic and plastic deformation, and we neglect (similarly to Vickers hardness measurement procedure) the formation of cracks near the impression.

Figure 2 presents the re-calculated results of cyclic nanoindentation for polycrystalline silicon (a) and coatings (b). The stresses of cracks initiation and propagation for coatings are significantly lower than those for silicon. The results for silicon are presented in Stress (MPa). It is most probable that steps of constant stresses for Si at loads of 30 mN and 50 mN correspond to the activation of dislocation sources or due to the joining of lateral cracks. The sharp drop in stress occurrences is due to their relaxation during coalescence and rapid propagation of brittle cracks. The noticeable difference in curves for 30, 40, and 50 mN can be explained by phase transformations that already developed at the initial moments of nanoindentation.

Silicon exhibits a phase transformation at pressures 9 to 16 GPa. Cubic diamond (Si-I) phase is transformed into the metallic one β-tin (Si-II) and accompanied by a densification (volume contraction) of about 20%. Experiments also indicate that the first phase formed from Si-II in 10–12 GPa, under slow decompression, was Si-XII (or R8-rhombohedral structure with eight atoms per unit cell), leading to a 9% expansion of material [23]. At further decompressing of Si, the degree of rhombohedral distortion gradually decreases and a mixture of Si-XII and Si-III phases (bcc 8 body-centred cubic structure with 16 atoms per unit cell) is produced, whereas the Si-XII remains at ambient pressure. Various investigations [20,21,22,23,24,25] have shown the probability of phase transformations in silicon during nanoindentation. One can assume that slip bands are arising during nanoindentation under load localisation, and their hindering provides peak stresses initiating phase transformations in local shear bands at a lower level in the range of 30–50 mN than was described in the literature [23].

One can see in Figure 1 and Figure 2 that opposed to brittle silicon, the single- and multilayer coatings behave as constant values of stresses and penetration depth achieved at indentation time from 5 to 35 s. This stage of cyclic nanoindentation is developing at a depth of about 100 nm. We can assume that dislocation sources are initiated at this time. Microplastic deformation in these coatings is developing without flashy growth of microcracks. For later impact testing times microcracks in the monolayer coating are nucleating and growing to the critical length. Stresses in the multilayer coating remain stable during a long enough time when microcracks grow, reaching their critical length. Then, microcracks grow without increasing the applied stresses propagating into the volume, which leads to stress relaxation. Microplastic deformation of a multilayer coating takes a much longer time and stops with gradual stress-decreasing as a result of the initiation and propagation of microcracks into the coating layers depth of about 280 nm corresponding to six TiAlCrSiYN/TiAlCrN bilayers. Formation of hills around indentations in the case of microplastic deformation and its absence at brittle cracking was previously shown on SEM images of imprints in [2,14] (Figure 3).

In accordance with the Mises principle [24], microplastic deformation can develop in neighbouring regions in the presence of compatible deformation. It can be assumed that a high number of interfaces in a multilayer coating should sharply reduce the plasticity due to the disarrangement in compatibility in the transfer of the dislocation slip between layers, but such a coating has a big margin of toughness compared with the monolayer coating. This is clearly seen when we compare indentation depth stabilisation from 7 s to 52 s of the impact. There are two reasons for this phenomenon. The first is that a good crystallographic conformity exists between the TiAlCrSiYN and TiAlCrN layers during their epitaxial growth. The second reason is that the initiation and growth of microcracks in each of the layers in the multilayer coating prevails over their transboundary propagation. At the same time, the mature substructure and the small thickness of several nanometre layers can significantly reduce peak Peierls stresses [26] and slow the growth of microcracks to their critical size. The stabilisation of the indentation depth and the constancy of stresses at long test times up to 300 s (Figure 1 and Figure 2) may be due to the strain hardening and the stabilization of coating plastic deformation at the selected indentation load. 

The monolayer coating has lower impact toughness and it is fractured with cascade formation of brittle cracks throughout the entire volume of the material. In the multilayer coating, microcracking is developing in several layers by sequential propagating of microcracks between layers. This process is energy-consuming with one comparing to simple crack branching in a monolayer. Obviously, during wear of coatings as in course of cyclic nanoindentation tests, mechanical load is accompanied by microplastic deformation, microcrack formation, and strain hardening.

It was of considerable interest to study the influence of initial stages of coating degradation on further ones. For this purpose, the indentation curves shown in Figure 1 and Figure 2 were subjected to fractal analysis.

We used the Hurst exponent as a tool measuring the long-term influence of previous events on subsequent ones developed during nanoimpact tests. To determine the Hurst exponent, nanoindentation curves presented in Figure 1 and Figure 2 for 300 s of testing were rescaled in short time intervals corresponding to individual jumps. For each series, the following was determined: the end of the time series, the average value of the penetration depth in each series, the difference between the maximum and minimum values (R), standard deviation (S), and number of measurements (pulses) (N). The Hurst parameter (H) was calculated using the following formula:H = log(R/S)/log(N/2),(3)

In this analysis three ranges of the Hurst exponent are considered. The H value of 0.5 corresponds to a stochastic uncorrelated process. In this case, the prior and posterior events are divorced from each other. The current process does not determine the following one. If 0 ≤ H < 0.5, the process is unstable similar to the “white noise” when a mean reversion is observed. A Hurst exponent in the range of 0.5 < H < 1 represents a sequence of events coupled in time. Here, events inherit from previous ones, and each previous event affects the next one. In terms of chaotic dynamics, the process sensitivity to initial conditions is observed in this case. Moreover, such long-term memory is preserved over the whole analysed time series. That is, short-term processes will affect further short-term processes, and long-term processes will affect further long-term processes. At H exceeding 0.5, the time series is a fractal and the series of self-similar processes are observed in it.

Figure 3 shows the cyclic variation of the Hurst exponent for nanoimpact tests of silicon. The blue colour corresponds to calculations based on Figure 1 depth-time plots; the red colour corresponds to the calculation based on Figure 2 stress-time. As one can see, weakly interdependent processes are observed only in the very initial period up to 15 s of nanoindentation at a load of 40 mN. This means that in the initial period of testing there is a close relationship between the microcracks initiation and their propagation. In all other cases during nanoindentation, the Hurst exponent value is less than 0.5, and degradation of silicon develops as a stochastic process. At the 40 mN load, the curves of the Hurst exponent calculated from the drop-in stresses or the increase in the depth of the crater, are in alignment, but at other loads the divergence of the curves becomes noticeable after 15 s. These regions are marked by ovals on the plots. Most likely this difference in Hurst exponents is observed because at this fatigue stage, the cascade of the lateral branching of cracks develops in silicon, and the real contact area between indenter and material distinguishes from what one expects for a given penetration depth. As the Hurst exponent value for silicon is on average about 0.5, its fatigue brittle fracture occurs as a stochastic process, when in the simplest case the branching of cracks does not relate to the number of their nucleation centres.

Comparing Figure 3 and Figure 4, one can see that the wear of both monolayer and multilayer coatings differs significantly from the degradation of silicon: Hurst exponents are staying above 0.5 up to 300 s of the fatigue tests.

Analysing the features of the plots in Figure 4, we can see that the Hurst exponent H fluctuates within 0.5 < H <0.66. For each coating the Hurst exponents coincide completely regardless of calculation method. This means that during fatigue tests the ductility of the coatings is high and brittle degradation of the material near the indenter print is not observed as distinguished from silicon. Each fluctuation peak H = F(t) signifies the accumulation of coherent structural transformations and their subsequent relaxation. Microplastic deformation and relaxation of accumulated micro-stresses occur at initial moments of the tests. The average level of the Hurst criterion is slightly higher for the multilayer coating than for the single layer one. This means that the processes of structural self-organization in both coatings are sufficiently high and that if the Hurst exponent is higher than 0.5 for a long series of observations, the long-term memory of events becomes significant.

Since the Hurst exponent is reliably higher than 0.5 during testing of up to 300 s, both coatings are damaged gradually due to the slow growth of previously initiated microcracks. Analysing Figure 2 and Figure 4, we can conclude that the multilayer coating has an increased low-cycle fatigue resistance compared to the monolayer coating. In the entire range of fatigue tests, microplastic deformations in the monolayer coating are developing for a short time at the very initial moment of loading. Subsequently, cracks are nucleating and propagating by the brittle fracture mechanism through nitride, spontaneously and at high speed without any interrelation. The multilayer coating is characterised by a big toughness margin. So, the fatigue in this coating develops with the presence of an incubation period when the initiation and propagation of microcracks is accompanied by microplastic deformation and relaxation of accumulated stresses without brittle degradation of coatings.

## 4. Conclusions

At the early stage of wear, microcracks under the influence of localised stresses are nucleating and propagating in the coatings. During the incubation period, the nucleation of cracks is accompanied by microplastic deformation, which significantly increases the stage of damage accumulation and influences on cracks branching in the volume and on the interfaces. In the monolayer coating the nucleation and growth of brittle microcracks occurs throughout the whole deformed volume. During low-cycle fatigue tests the structural self-organization is quite high in both multilayer TiAlCrSiYN/TiAlCrN and monolayer TiAlCrSiYN coatings. In this case, the multilayer coating is characterised by larger toughness. In these coatings, fatigue structural changes are interrelated in contrast to silicon where the low-cycle fatigue develops as a stochastic accumulation of imperfections.

## Figures and Tables

**Figure 1 nanomaterials-12-01312-f001:**
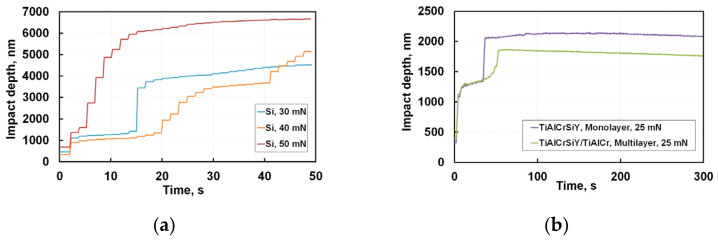
Short-term indenter penetration depth at 30–50 mN, 50 s for Si reference sample (**a**) and for monolayer and multilayer coatings at 25 mN, 300 s (**b**).

**Figure 2 nanomaterials-12-01312-f002:**
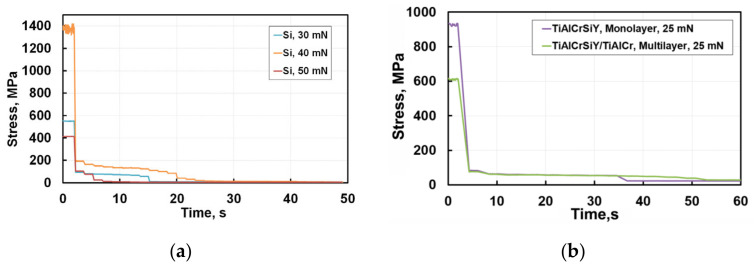
Dependence of stress on the cyclic nanoindentation time at 30–50 mN, 70 s for Si reference (**a**) and at 25 mN, 70 s for monolayer and multilayer coatings (**b**).

**Figure 3 nanomaterials-12-01312-f003:**
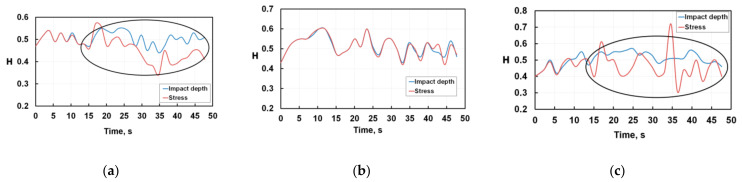
Cyclic variations of Hurst exponents for fatigue tests of silicon at 30 mN (**a**), 40 mN (**b**), and 50 mN (**c**) loads.

**Figure 4 nanomaterials-12-01312-f004:**
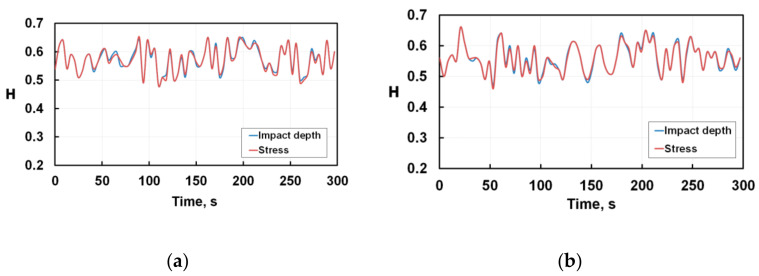
Cyclic variations of Hurst exponents for nanoindentation tests of monolayer (**a**) and multilayer (**b**) coatings.

## Data Availability

Not applicable.

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
