# Peer review of "Damage Accumulation Phenomena in Multilayer (TiAlCrSiY)N/(TiAlCr)N, Monolayer (TiAlCrSiY)N Coatings and Silicon upon Deformation by Cyclic Nanoindentation"

_nanomaterials, 2022, doi:10.3390/nano12081312_

Round 1

Reviewer 1 Report

The authors used cyclic nanoindentation technique to evaluate the fatigue property of a novel multilayer coating that was used in a cutting tool. The indentation results suggest accumulation of damage by microplastic deformation in the multilayer structure. It is suggested to give further experimental evidences (like using SEM/FIB to have a detailed observation of the indent cross-sections) on the plasticity of the multilayer after indentation, to support this conclusion.  In addition, there are still some typo/grammatical errors in the whole manuscript. 

Author Response

Thank you very much for your comments..

 We added references [14] and [28] where SEM data on microplastic deformation around impact craters were presented. There was shown build-up around the indentations and its absence in the case of brittle fracture, as well as a pattern of crack propagation between layers.

The text of the article has been re-edited by us and all grammatical errors have been corrected

Reviewer 2 Report

In this paper, an original approach was used to understand the low-cycle fatigue properties of different layers. Studied samples were prepared using by PVD technique. It should be emphasized that these layers are used in the tool industry to increase the lifetime of the cutting tools, proving the applicability of this work. Therefore, I appreciate the effort of the authors to present this research and even more to use a comprehensive experimental methodology that helps to grasp the overall concept of the work. 
Speaking about the scientific quality of the paper, I appreciate using the nanoindentation technique and proving the possibility to study low-cycle fatigue properties. Reported investigations give deep insight into the chemical and psychical mechanisms of the layers. Due to the importance of the presented research, the reported study should be published in the Nanomaterials journal. However, before that, I recommend performing a minor revision of the article to clear some aspects of the manuscript. More detailed suggestions are listed in the following points: 

No SEM image before and after indentation is provided. 
Discussion about cracks development should be enriched into the structural analysis.
L-D curve is necessary to be shown. Pop in behavior is mentioned. Did the authors record that too? 
How can one prove that phase transition occurred, XRD, Raman (in Si), at least data from the literature should be provided? 
The authors discuss the deformed zone's plastically deformed region and microcrack development. This needs to be supported by some SEM or TEM data. It will significantly improve the quality of the paper. 

In conclusion, the presented paper is in line with the current trend of the life improvement of cutting tools. Therefore, it can be considered necessary for engineers worldwide. However, before the publishing process starts, the authors should work a little harder to present their results more clearly (especially the structural part of the article that needs to be improved). 

Author Response

Thank you very much for your valuable comments.

– «No SEM image before and after indentation is provided. Discussion about cracks development should be enriched into the structural analysis. The authors discuss the deformed zone's plastically deformed region and microcrack development. This needs to be supported by some SEM or TEM data. It will significantly improve the quality of the paper».

 We added the text with references [14] and [28] to our articles where SEM data on microplastic deformation around impact crater were presented: “Formation of hills around indentations in the case of microplastic deformation and its absence at brittle cracking was previously shown on SEM images of imprints in [14] (Fig. 7) and [28] (Fig. 3).”.

– «L-D curve is necessary to be shown. Pop in behavior is mentioned. Did the authors record that too?»

In this work, we have not explored single-cycle force-displacement load and unload diagrams. These results were presented in our previous paper [G. S. Fox-Rabinovich,J. L. Endrino, M. H. Agguire, B. D. Beake, S. C. Veldhuis, A. I. Kovalev, I. S. Gershman, K. Yamamoto, Y. Losset, D. L. Wainstein, and A. Rashkovskiy. Mechanism of adaptability for the nano-structured TiAlCrSiYN-based hard physical vapor deposition coatings under extreme frictional conditions. JOURNAL OF APPLIED PHYSICS 111, 064306 (2012).]

We didn't want to increase the amount of self-citation. This work is a development of previously published studies.

«How can one prove that phase transition occurred, XRD, Raman (in Si), at least data from the literature should be provided?»

We added to the discussion references to phase transformations in silicon during cyclic nanoindentation [20, 21, 22, 25, 29, 30]. A lot of data on phase transformations in silicon during nanoindentation have been published. We consider it to be inappropriate to repeat these well-known studies.

Round 2

Reviewer 1 Report

The quality of the manuscript is improved after correction.